# Analytical Model for Temperature Prediction in Milling AISI D2 with Minimum Quantity Lubrication

Linger Cai [1,*], Yixuan Feng [1], Yu-Ting Lu [2], Yu-Fu Lin [2], Tsung-Pin Hung [3], Fu-Chuan Hsu [2] and Steven Y. Liang [1,*]

[1] Georgia Institute of Technology, Woodruff School of Mechanical Engineering, 801 Ferst Drive, Atlanta, GA 30332, USA; yfeng82@gatech.edu

[2] Metal Industries Research and Development Centre (MIRDC), Kaohsiung 81160, Taiwan; lyting@mail.mirdc.org.tw (Y.-T.L.); yufulin@mail.mirdc.org.tw (Y.-F.L.); fchsu@mail.mirdc.org.tw (F.-C.H.)

[3] Center for Environmental Toxin and Emerging-Contaminant Research, Super Micro Mass Research & Technology Center, Department of Mechanical Engineering, Cheng Shiu University, Kaohsiung 83347, Taiwan; tphung@gcloud.csu.edu.tw

\* Correspondence: lingercai@gatech.edu (L.C.); steven.liang@me.gatech.edu (S.Y.L.)

**Abstract:** Milling with minimum quantity lubrication (MQL) is now a commonly used machining technique in industry. The application of the MQL significantly reduces the temperature on the machined surface, while the cost of the lubricants is limited and the pollution caused by the lubricants is better controlled. However, the fast prediction of the milling temperature during the process has not been well developed. This paper proposes an analytical model for milling temperature prediction at the workpiece flank surface with MQL application. Based on the modified orthogonal cutting model and boundary layer lubrication effect, the proposed model takes in the process parameters and can generate the temperature profile at the workpiece surface within 1 min. The model is validated with experimental data in milling AISI D2 steel. With an average absolute error of 10.38%, the proposed model provides a reasonable temperature prediction compared to the experimental results. Based on the proposed model, this paper also investigates the effect of different cutting parameters on the cutting temperature. It is found that the application of the MQL decreases the temperature at the cutting zone, especially at the flank surface of the workpiece, which is due to the heat loss led by air-oil flow.

**Keywords:** cutting temperature; analytical modeling; Johnson–Cook; minimum quantity lubrication

## 1. Introduction

The minimum quantity lubrication (MQL) technique is widely used in industries because of its superiority in environmental protection as well as cost efficiency. Compared to dry machining and flood machining, the application of MQL reduces cutting forces [1], delivers better surface finishes [2], and leads to less tool wear [3]. During the application of MQL, a small amount of lubricant, usually between 10 and 100 mL/h, is sprayed onto the cutting zone. These applied lubricants lower the friction during the cutting process [4], which further reduce the temperature around the cutting area. In the conventional cutting process [1–3], the high temperature generated in the machining process often causes impaired accuracy, shorter tool life, as well as the diminished surface integrity [5]. Therefore, to achieve optimal cutting outcomes, it is valuable to study the temperature around the cutting during the machining process with MQL application.

Most researchers have studied the temperature by experiments and measurements. Le Coz et al. [6] presented several temperature measurement techniques and admitted the difficulties and deviations for temperature measurement accuracy. They investigated the temperature during drilling operation with a Ti6Al4V alloy with the application of MQL. However, their result is much higher those measured by Zeilmann and Weingaertner [7]

with similar conditions. This difference reveals the difficulty of temperature measurement during the machining process. Hadad et al. [8] reported the temperature in grinding 100Cr6 steel with $Al_2O_3$ and CBN wheels for dry, MQL, and fluid environments. Under the MQL condition, Hadad investigated several different combinations of MQL coolant lubricants and delivery methods. All MQL combinations showed temperature reductions compared to those measured under dry conditions. Yet, even though MQL delivered good lubrication, it did not meet the temperature reduction level as flood lubrication did. Li et al. [9] investigated the effects of different MQL base oils on grinding. The high-temperature nickel base alloy, GH4169, was used. Their results showed the superiority of the palm oil. As a base oil, when applied on grinding operation, the palm oil led to the lowest temperature and the highest energy ratio coefficient. Qin et al. [10] conducted experiments on turning the TC11 alloy with MQL. The experiments were performed with uncoated carbide inserts and $Al_2O_3$/TiAlN-coated tools. Their results showed a great amount of temperature drop due to the application of MQL compared to the dry condition. This research has proven that the application of MQL can effectively reduce heat generation in various machining operations.

While the heat reduction effect of MQL has been well-recognized, more studies have been conducted to determine the optimal parameters recently. Salur et al. [11] performed end-milling experiments on AISI 1040 steel with further ANOVA and Taguchi signal to noise analysis. In addition to the reduction in cutting temperature, their results also indicated that the application of MQL provided less tool wear and lower power consumption. According to their study, the combination of high feed rate and low cutting speed ensured lower temperature under MQL. Dubey et al. [12] performed turning experiments on AISI 304 steel, and found a significant temperature reduction, force reduction, and surface smoothness. After the application of several multicriteria decision-making techniques (MCDM), they determined the optimal set of process parameters to be a cutting velocity of 90 m/min, feed rate of 0.08 mm/min, depth of cut of 0.6 mm, and nanoparticle concentration of 1.5%.

Aside from the experimental methods and statistical analysis, the modeling methods are also used to determine the optimal process conditions. However, the modeling and prediction of machining temperature under MQL were much less found in literature. One of the reasons is that the interpretation of the heat reduction effect of MQL is difficult to derive. In theory, when MQL is applied, the friction coefficient at the contact area is lowered, which usually results in the decrease in the cutting force and changes in the flow stress. The influencing mechanism of the friction coefficient has been evaluated by several researchers. Based on the strain gradient and geometry and kinematics analysis, Yang et al. [13] proposed a minimum chip thickness model for grinding operation. In their model, the lubrication condition was represented by the friction angle. Their model showed that a larger friction angle leads to a smaller minimum chip thickness. In the context of orthogonal cutting, Zhang et al. [14] investigated the influence of limiting shear stress at the tool–chip interface in the case of Ti-6Al-4V. In this study, they found that the friction coefficient is affected, which led to their further analysis of the influence of friction coefficient on the chip morphology. It was found that the friction coefficient significantly affected the temperature distribution on the tool–chip interface. However, in their analysis, the change in friction coefficient did not lead to the cutting force change, because more energy was transferred into heat and softened the materials.

Even though the modeling methods were applied, most of the work carried out in temperature prediction were based on numerical methods. Morgan et al. [15] used the thermal model reported and summarized by Rowe [16] to predict the temperature in grinding operation. They further applied the dry model with the expected higher temperature. Biermann et al. [17] presented a finite element simulation for the thermal behavior during deep hole drilling operation. They considered the process with temporal and spatial discretization, material removal effects, and additional heat sources. Their model was then validated with experimental data, which led to reasonable comparison results. Kaynak et al. [18] investigated turning operation with Ti-5553. They integrated

the orthogonal cutting model with the finite element method. With this new model, they predicted the cutting temperature distribution with the influence of cryogenic, MQL, and high-pressure coolant supplies. The numerical methods are easier to apply but also have the disadvantage of high computational cost.

The analytical method, on the other hand, eliminates the interaction procedure in numerical methods and has a comparably higher computational efficiency. However, it is less introduced in the literature as the derivation of the analytical solution is more case-to-case and less generalized. Hadad et al. and Sadeghi [19] proposed an analytical thermal model for the grinding process. Based on Hanna's model [20], Hadad et al. further included the scale of the workpiece-tool combinations, grain-workpiece contact length, and the heat transfer due to MQL. Their model was then validated by experiments on the grinding of 100Cr6 steel. Improved from Li and Liang's model [21], Ji et al. [22] predicted the machining temperature during the turning process to AISI 9310. This model was then validated by experimental data and provided a reasonable result.

As Le Coz et al. [6] stated, little research has been conducted to understand the temperature during milling operation due to the difficulty in the measurements. Studies on the temperature prediction in the milling process with the MQL technique remain even fewer. This paper aims to fill that gap and provide an analytical model for temperature prediction in the milling process with MQL application. The proposed model is based on the chip formation orthogonal cutting model [23] with consideration of material properties modeled by the Johnson–Cook model. The 3D milling operation is transferred into 2D orthogonal cutting based upon the orthogonal equivalent representation proven effective [24]. The effects of MQL are considered with the boundary layer lubrication effect analysis together with three heat sources, namely the primary, the secondary, and the heat loss region considered as in Ji's study [25].

The proposed model is then validated with experimental data with the milling of AISI D2. This material is chosen for its wide usage [26]. While AISI D2 can be used as a high-efficiency cutting tool, its superior hardness and toughness also make it difficult to be machined. Thus, the development of a nonconventional machining process for the material is necessary [27]. After the model is calibrated and validated, a sensitivity analysis is performed to provide a better understanding of the effects of cutting parameters on the temperature.

## 2. Analytical Model for Temperature Prediction in Milling

### 2.1. Instantaneously Transferring End-Milling Condition into Orthogonal Cutting

In the proposed model, at every moment when the tool edge cuts the material, it is considered as an orthogonal cutting condition. The average depth of cut $\bar{t}_c$ is calculated as [24]

$$\bar{t}_c = \frac{1}{2}\frac{V_f}{RPM'} \tag{1}$$

where $V_f$ is the feed rate and the instantaneous equivalent depth of cut at each tool rotation angle is $t_c(\psi) = \sqrt{2} \times \bar{t}_c \times sin(\psi)$, where $\psi = 2\pi R \times RPM \times t$ at given time $t$.

The definition of the side cutting edge angle $C_s{}^*$ at given time $t$ is

$$C_s{}^* = C_s + \eta_c, \tag{2}$$

where $C_s$ is the tool side cutting angle, and the chip flow angle $\eta_c$ is calculated based on tool geometry and cutting parameters [17].

The equivalent chip flow angle $\eta_c^*$ and the equivalent inclination angle $i^*$ are

$$\eta_c^* = i^* = \arcsin(\cos\eta_0\sin i - \sin\eta_0\sin\alpha\cos i), \tag{3}$$

where $i$ is the inclination angle, $\alpha$ is the rake angle, and $\eta_0$ is defined as

$$\eta_0 = \arccos\left(\frac{\sec i - \tan i \tan\eta_c \tan\alpha}{\sqrt{(\tan i - \tan\eta_c \tan\alpha \sec i)^2 + \sec^2\eta_c}}\right)., \tag{4}$$

The equivalent rake angle is

$$\alpha^* = \arcsin\left(\frac{\sec\eta_0 \sin i - \sin i^*}{\tan\eta_0 \cos i^*}\right), \tag{5}$$

For the equivalent orthogonal cutting, the equivalent cutting depth $t_c^*$ is

$$t_c^* = t_c(\psi) \times \cos C_s^*, \tag{6}$$

The cutting width in orthogonal cutting is related to the axial depth of milling as

$$w^* = \frac{d}{\cos(C_s^*)}, \tag{7}$$

The equivalent cutting speed is a function of the rotation angle as

$$V(\phi) = \sqrt{V_f^2 + V_r^2 + 2V_f V_r \cos\psi}., \tag{8}$$

where $V_r = 2\pi R \times RPM$ is the rotation speed and $R$ is the tool radius.

### 2.2. Friction Coefficient Calculation under Minimum Quantity Lubrication Condition

In this model, the lubricant is assumed to be applied between the flank surface of the tool and the workpiece surface by a separated nozzle. It is assumed that the lubricant forms a thin film at the surface of the workpiece where the boundary lubrication is applied. The friction coefficient is then modified based on the boundary layer lubrication model, which is further applied into Oxley's orthogonal cutting model [22].

In the proposed model, the normal load $N$ and the friction force $F$ at the boundary are defined as

$$N = p_m A_{ms} + p_b A_{bs}, \tag{9}$$

$$F = s_m A_{ms} + s_b A_{bs} \tag{10}$$

where $p_m$, $s_m$ and $p_b$, $s_b$ are the yield pressure and shear strength at the metallic contact area and the adsorbed lubricant film contact area, respectively. $A_{ms}$, the metallic contact area, is defined as

$$A_{ms} = \frac{\pi R n_0 D^2 a_s^3}{6 H_{\max}^2}, \tag{11}$$

In addition, the adsorbed lubricant film contact area $A_{bs}$ is defined as

$$A_{bs} = \frac{\pi R n_0 D^2 \left\{(a_s + t_b)^3 - a_s^3\right\}}{6 H_{\max}^2}, \tag{12}$$

where $R$ is the asperity tip radius, $n_0$ is the total asperity number, $D$ is the inclination distribution function, $a_s$ is the approach of two surfaces, $H_{\max}$ is the asperity height distribution, and $t_b$ is the effective adsorbed lubrication film thickness.

After the above parameters are determined, the friction coefficient is defined as

$$\mu = \frac{F}{N} = \frac{s_m A_{ms} + s_b A_{bs}}{p_m A_{ms} + p_b A_{bs}}, \tag{13}$$

Now, define

$$C_1 = \frac{s_m}{p_m}, C_2 = \frac{p_b}{p_m}, C_3 = \frac{s_b}{p_b}, \tag{14}$$

Then, the friction coefficient is

$$\mu = \frac{C_1 A_{ms} + C_2 C_3 A_{bs}}{A_{ms} + C_2 A_{bs}}, \tag{15}$$

$a_s$ can be solved from

$$a_s^3 + 3C_2 t_b a_s^2 + 3C_2 t_b^2 a_s + \left( C_2 t_b^3 - \frac{N}{p_m Q} \right) = 0, \tag{16}$$

where

$$Q = \frac{\pi R n_0 D^2}{6 H_{\max}^2}, \tag{17}$$

Then,

$$\mu = \frac{C_1 a_s^3 + C_2 C_3 \left\{ (a_s + t_b)^3 - a_s^3 \right\}}{a_s^3 + C_2 \left\{ (a_s + t_b)^3 - a_s^3 \right\}}, \tag{18}$$

The variable $N$ here is the normal load at the surface, which is estimated based on the modified orthogonal cutting model without lubrication applied. In this paper, $t_b$ is mostly affected by the flow rate of the lubricant. It is assumed to be linearly correlated with the flow rate to curtain level depending on the specific cutting condition. The case where $t_b$ equals to zero corresponds to the dry milling condition. Based on Equation (18), $C_1$ represents the friction coefficient under dry conditions. The variables $C_2$ and $C_3$ are based on the properties of the applied lubricant, which is calibrated by one experimental force point.

The calculated friction coefficient is then transferred into friction angle in the orthogonal cutting model with

$$\lambda = \arctan(\mu), \tag{19}$$

where $\lambda$ is taken as the friction angle in the orthogonal cutting model. Then, the angle between the force and the shear plane $\phi$ is given by

$$\theta = \phi + \lambda - \alpha, \tag{20}$$

where $\phi$ is the shear angle and $\alpha$ is the rake angle.

### 2.3. Temperature Prediction in Orthogonal Cutting with Minimum Quantity Lubrication

After the 3D milling motion is transferred into 2D orthogonal cutting (Section 2.1) and the friction angle is modified based on the lubrication condition (Section 2.2), the proposed model then considers the effect of the MQL. In the proposed model, the lubricants are assumed to be applied by an exterior nozzle and the lubricant mainly affects the tool–workpiece interface. The application of MQL leads two main effects: the friction coefficient change at the tool–workpiece interface and the heat loss at the flank tool surface. In that case, the total heat change in the milling process is integrated into three regions, as illustrated in Figure 1.

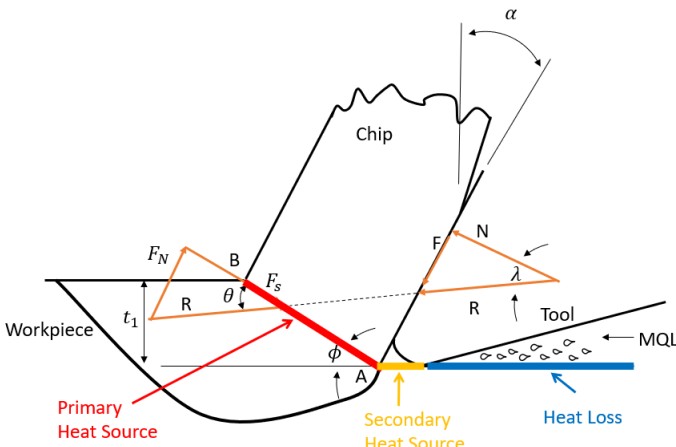

**Figure 1.** Illustration of considered heat change regions.

The first region is the shear zone, which is also called the primary heat source. The heat generated here is caused by shear deformation of the workpiece. The second region is located on the tool–workpiece interface. It is also called the secondary heat source. The heat generated here results from the rubbing effect between the tool and the workpiece. In both regions, heat is generated. However, in the third region, which is at the frank tool surface where the lubricant is applied, the temperature drops because of the cooling effect of the applied lubricant.

To calculate the temperature changes from each source, the cutting force $F_c$ and the radial force $F_t$ at this instance (at this rotation angle $\psi$) are calculated based on a modified Oxley's model. As this model has been extensively used in the area, this paper does not repeat its formulation.

Figure 2 gives the illustration of the primary heat source, which is also the shear plane of the chip. The temperature rise in the workpiece from the primary heat source is calculated as follows:

$$
\begin{aligned}
T_{primary}\,(X,Z) = \ & \frac{q_{shear}}{2\pi k_{wk}} \int_0^{L_{AB}} e^{-\frac{(X-l_i\cos\phi)V}{2a_{wk}}} \left\{ K_0 \left[ \frac{V}{2a_{wk}} \sqrt{(X-l_i\cos\phi)^2 + (Z+l_i\sin\phi)^2} \right] \right. \\
& \left. + K_0 \left[ \frac{V}{2a_{wk}} \sqrt{(X-l_i\cos\phi)^2 + (2t_1-l_i\sin\phi+Z)^2} \right] \right\} dl_i
\end{aligned}
\tag{21}
$$

where $L_{AB}$ is the length of the shear plane, given by $L_{AB} = \frac{t_1}{\sin\phi}$. $\phi$ is the shear angle and $t_1$ is the undeformed chip thickness. The variable $V$ is the cutting speed. $a_{wk}$ is the thermal diffusivity of the workpiece. $k_{wk}$ is the thermal conductivity of the workpiece. $K_0$ is the modified Bessel function. $q_{shear}$ is the heat source density in the shear zone, given as follows:

$$
q_{shear} = \frac{(F_c\cos\phi - F_t\sin\phi)(V\cos\alpha^*/\cos(\phi-\alpha^*))}{t_c^* * w^* * \csc\phi},
\tag{22}
$$

where $\alpha^*$ is the equivalent rake angle calculated by Equation (5), $t_c^*$ is the equivalent cutting depth calculated by Equation (6), and $w^*$ is the equivalent cutting width calculated by Equation (7).

Figure 3 shows the illustration of the secondary hear source. The temperature rise in the secondary heat source here is calculated as follows:

$$
T_{Secondary}\,(X,Z) = \frac{q_{rub}}{\pi k_{wk}} \int_0^{CA} \gamma e^{-\frac{(X-x_i)V}{2a_{wk}}} \left\{ K_0 \left[ \frac{V}{2a_{wk}} \sqrt{(X+x_i)^2 + Z^2} \right] \right\} dx_i
\tag{23}
$$

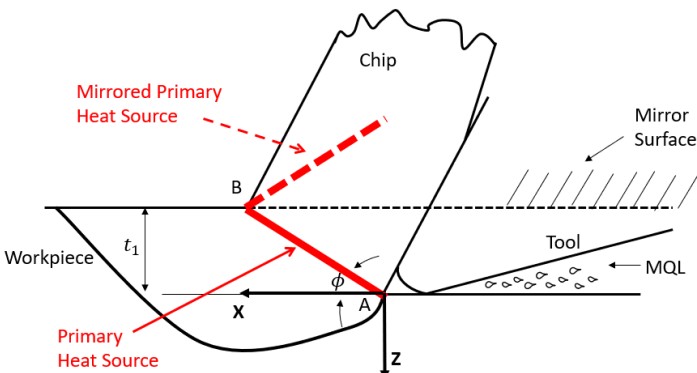

**Figure 2.** Illustration of the primary heat source.

The variable $CA$ is the contact length calculated based on the slip-line model [28], which is calculated in the process. $\gamma$ is the heat distribution coefficient based on material properties of the workpiece and the cutting tool, which is defined as follows:

$$\gamma = \frac{\sqrt{k_{wk}\rho_{wk}C_{wk}}}{\sqrt{k_{wk}\rho_{wk}C_{wk}} + \sqrt{k_t\rho_t C_t}},\tag{24}$$

where the variable $k_{wk}$ is the thermal conductivity of the workpiece, $\rho_{wk}$ is the density of the workpiece, and $C_{wk}$ is the thermal capacity of the workpiece. $k_t, \rho_t, C_t$ are the same properties of the cutting tool, respectively. $q_{rub}$ is the heat source density in the rubbing zone, given as follows:

$$q_{rub} = \frac{P_{cut}\,V}{w^* * CA},\tag{25}$$

The variable $P_{cut}$ is the plowing force calculated based on Waldorf's study [28].

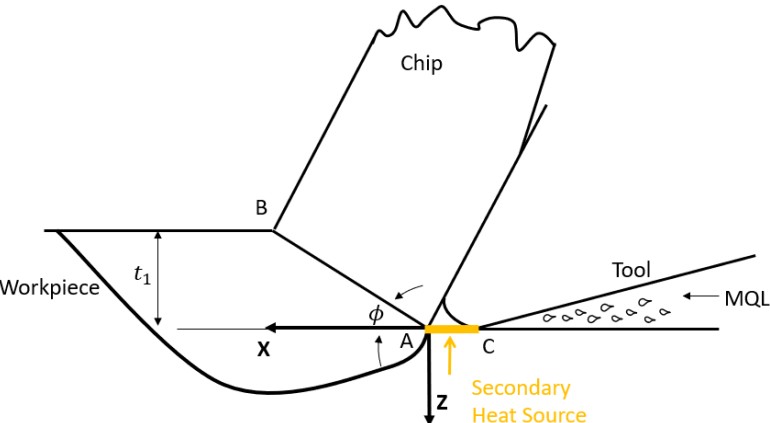

**Figure 3.** Illustration of the secondary heat source.

Figure 4 illustrates the heat loss in the third region. The generated heat loss in this region is calculated as follows:

$$T_{loss}\,(X,Z) = \frac{q_{loss}}{\pi k_{wk}}\int_0^{L_t} e^{-\frac{(-X+L_{AB}\cos\phi+x_i)V}{2a_{wk}}}\left\{K_0\left[\frac{V}{2a_{wk}}\sqrt{(X-L_{AB}\cos\phi-x_i)^2+(Z+t_1)^2}\right]\right\}dx_i\tag{26}$$

where $q_{loss}$ is the heat loss intensity due to air-oil flow, given as follows:

$$q_{loss} = \overline{h}\left(T_{flank} - T_w\right),\tag{27}$$

where $T_{flank}$ is the average tool flank face temperature and $T_w$ is the room temperature. $\bar{h}$ is the average heat transfer coefficient, which is estimated by the Nusselt number as follows:

$$\bar{h} = 0.664 \text{P}_r^{1/3} * Re^{1/2} * k_{air}/L_{eff}, \tag{28}$$

where Pr is the Prandtl number, Re is the Reynolds number, $k_{air}$ is the thermal conductivity of the air, and $L_{eff}$ is the effective lubricated length, which equals $L_t$.

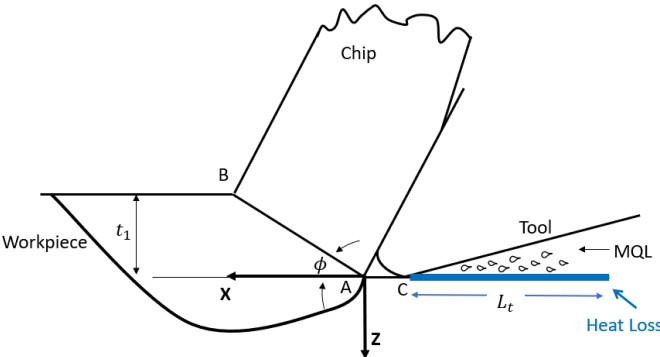

**Figure 4.** Illustration of the heat loss region.

The final temperature of point (X,Z) on the workpiece is then expressed as follows:

$$T_{wk}(X, Z) = T_{primary}(X, Z) + T_{secondary}(X, Z) - T_{loss}(X, Z) + T_{room}, \tag{29}$$

*2.4. Temperature Prediction Flow Chart*

Figure 5 shows the flow chart of the model calculation process. Based on the input cutting conditions, an estimated cutting force is first calculated based on the orthogonal cutting model. The preliminary force result is then put into the boundary lubrication effect model to calculate the modified friction coefficient. The updated friction coefficient is then transferred into the friction angle based on Equation (19), which is then put back into the modified orthogonal cutting model for the prediction of the temperature field.

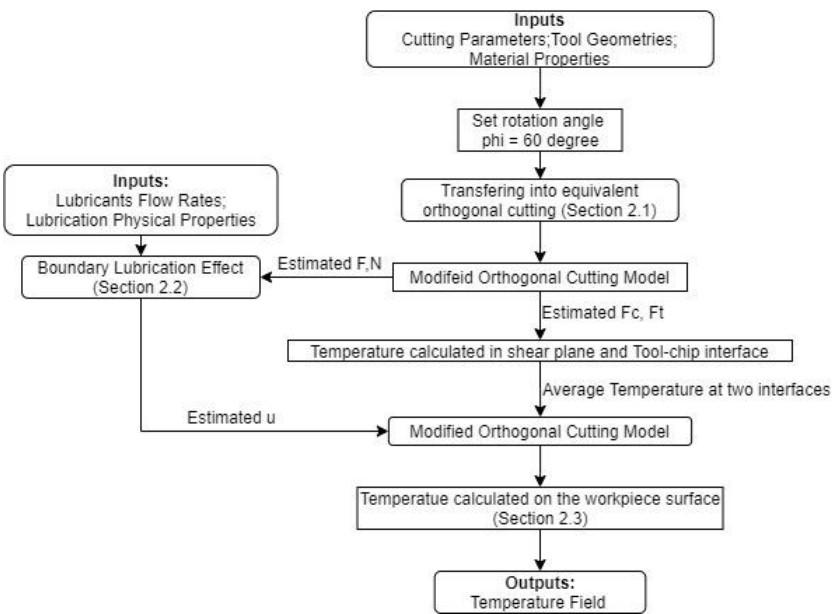

**Figure 5.** Flow chart of the temperature calculation.

## 3. Experimental Data Validation

The proposed model was then validated with published data. Khan et al. [29] performed experiments on Chromium-based D2 steel (AISI D2 steel), which had a hardness of 60 HRC. The chemical composition of the workpiece is given in Table 1, and the used sample had dimensions of $100 \times 50 \times 8$ mm$^3$. The experimental setup is shown in Table 2. Figure 6 shows the experimental setup and the geometry of the cutting tool [29]. The real-time temperature was measured by using an infrared thermometer Raytek Raynger MX4. The laser was aimed at the workpiece–tool interface. As Khan stated, the device measured the real-time value temperature with an accuracy of $\pm 1\,^\circ$C.

**Table 1.** Chemical composition of the workpiece [29].

| Element | C | Si | Mn | Cr | Mo | V | P | S | NI | Fe |
|---|---|---|---|---|---|---|---|---|---|---|
| %Weight | 1.56 | 0.30 | 0.4 | 11.9 | 0.78 | 0.80 | 0.023 | 0.015 | 0.05 | Balance |

**Table 2.** Experimental Setup [29].

| | |
|---|---|
| **Machine Tool** | Micron UCP 710 |
| **Work Materials** | AISI D2 Steel (Dimension: $100 \times 50 \times 8$ mm$^3$) |
| **Hardness** | 60 HRC |
| **Cutting Tool** | SECO-made Tungsten Carbide Inserts |
| **Process Parameters** | |
| **Feed Rate** | 100, 150, 200 mm/min |
| **Depth of Cut** | 0.2, 0.5, 0.8 mm |
| **Flow Rate** | 200, 300, 400 mL/h |
| **Cutting Speed** | 30 m/min (constant) |
| **MQL Fluid** | Distilled Water |
| **MQL Application** | Nozzle fixed at 45°, 15 mm away |

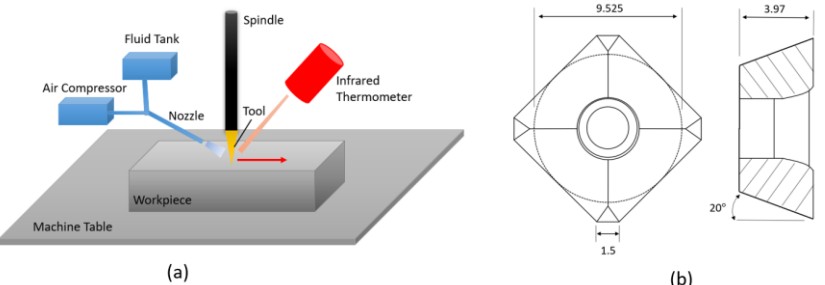

**Figure 6.** The experimental setup (**a**) and geometry of the cutting tool (**b**) [29].

In the proposed model, the workpiece material was modeled with the Johnson–Cool material model for the flow stress calculation. Table 3 gives the material properties, as well as Johnson–Cool parameters for AISI D2 tool steel.

Matlab R2018b was used for the temperature calculation. The computer used had 16 GB of memory and a 2.8 GHz CPU. The calculation time for each case was less than 1 min. A calibration process was performed using the experimental data from one test group 1 for the assumed variables, so it is not presented in the later table. The calibrated and assumed model parameters are listed in Table 4. Table 5 shows the results from the experiments and corresponding predicted temperature results. A rotation angle $\Psi$ of 60° was used for the prediction. The influences of the rotation angle are discussed in a later section. Because the temperature measurement laser was pointed at the workpiece–tool interface, it is assumed that the measured temperature was the maximum temperature on the workpiece. The predicted maximum temperature was used for comparison as well. The maximum error was 23.44% as in group 8. The minimum error was 0.7% in group 12. The average error was 10.38%. The proposed model tended to have a larger error

compared to the experimental data when a higher feed rate and higher lubricant flow rate were combined, such as group 8 with an error of 23.44% and group 13 with an error of 19.18%. The predicted temperature was higher than the measured ones in both cases. As mentioned previously in Section 2.3, the proposed model assumed that the lubrication only occurs at the tool–workpiece area and the heat loss is only calculated in this area as well. However, in the practical situation, the lubrication could happen at other areas of the machining process as well, especially when a higher feed rate is applied. The higher feed rate may lead to a higher air flow around the cutting zone, which leads to more heat loss that is not full considered in the proposed model. This could be the reason for the higher temperature predicted by the proposed model.

**Table 3.** Properties of AISI D2 Steel [30].

| Parameters | Number |
|:---:|:---:|
| A (MPa) | 1766 |
| B (MPa) | 904 |
| C | 0.012 |
| m | 3.38 |
| n | 0.312 |
| Density (g/cm$^3$) | 7.75 |
| Young's Modulus (GPa) | 180 |
| Thermal Conductivity (W/mK) | 21 |
| Melting Temperature (°C) | 2590 |

**Table 4.** Calibrated model parameters.

| Parameter | $C_2$ | $H_{max}(\mu m)$ | $R(\mu m)$ | $v(mm^2/s)$ | $\rho(g/mm^3)$ | $D$ |
|:---:|:---:|:---:|:---:|:---:|:---:|:---:|
| Value | 0.3 | 20 | 5 | 10 | 0.89 | 1.5 |

**Table 5.** Experimental measurements and predicted data with AISI [29].

| Group Number | Feed Rate (mm/min) | Depth of Cut (mm) | Flow Rate (mL/h) | Temperature Measured (°C) | Temperature Predicted (°C) | Absolute Error Percentage (%) |
|:---:|:---:|:---:|:---:|:---:|:---:|:---:|
| 2 | 200 | 0.2 | 300 | 135 | 148 | 9.63 |
| 3 | 100 | 0.8 | 300 | 124 | 132 | 6.45 |
| 4 | 200 | 0.8 | 300 | 189 | 170 | 10.05 |
| 5 | 100 | 0.5 | 200 | 129 | 143 | 10.85 |
| 6 | 200 | 0.5 | 200 | 184 | 191 | 3.80 |
| 7 | 100 | 0.5 | 400 | 108 | 104 | 3.70 |
| 8 | 200 | 0.5 | 400 | 128 | 158 | 23.44 |
| 9 | 150 | 0.2 | 200 | 165 | 138 | 16.36 |
| 10 | 150 | 0.8 | 200 | 183 | 203 | 10.93 |
| 11 | 150 | 0.2 | 400 | 116 | 127 | 9.48 |
| 12 | 150 | 0.8 | 400 | 143 | 144 | 0.70 |
| 13 | 150 | 0.5 | 300 | 146 | 174 | 19.18 |

Aside for the limitation in the higher-feed-rate higher-flow-rate case, the proposed model gave a reasonable temperature prediction result compared to the experimental data within a short time. To gain more understanding of the effects of the parameters applied, further analysis is performed in the following section.

## 4. Model Analysis

### 4.1. Effect of the Application of MQL

As mentioned before, the application of lubricants has mainly two effects: lowers the friction coefficient and increases the heat transfer around the cutting zone. Both result in a theoretical decrease in temperature. Figure 7 shows the predicted temperature field on the workpiece surface around the cutting zone with a feed rate of 200 mm/min, depth of cut of 0.5 mm, and MQL of 200 mL/h. Figure 8 shows the predicted temperature field using the same cutting parameters but without MQL applied. In the MQL case (Figure 7), the peak temperature in this case is 191.6 °C, which occurs at the point where the workpiece and the tool contact. The maximum temperature occurs at the same point for the dry case, but the absolute temperature reaches 260.3 °C, which is 35.8% higher than that of the MQL case. The lowest temperature in the MQL case occurs at the flank surface where the lubricant is assumed to be applied. The value of the lowest predicted temperature is 9.04 °C. At this point, the thermal loss caused by lubricants application is at the maximum rate, which leads to a significantly low temperature. In the dry case, there is no rapid temperature drop, because no extra thermal loss is applied and the temperature decreases smoothly. This smoothness is better shown in Figure 9, which presents the temperature at the workpiece surface. It can be seen that MQL provides a significantly lower temperature than the dry condition at both the tool tip and flank surface. In MQL, a faster temperature convergence to room temperature is also found, which is consistent with the expectation. From a physical point of view, the application of MQL causes an extra heat dispersion effect due to the higher heat coefficient and lower friction coefficient. Because of that, a larger temperature decrease rate is observed at the workpiece surface. At the lubricant application point, the temperature reaches the lowest point due to high heat loss. $T_{loss}$ is dominant at this location. Yet, for points behind the application point (away from the cutting zone), the heat loss is not as strong and the heat generated at the cutting zone is still effective. This leads to a small temperature rise and then generally returns to room temperature.

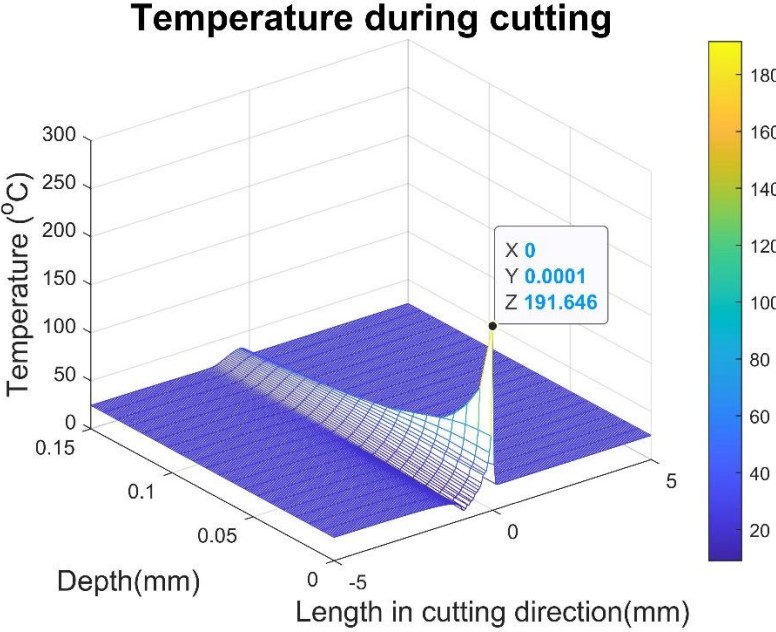

**Figure 7.** Predicted temperature field with feed rate of 200 mm/min, depth of cut of 0.5 mm, and flow rate of 200 mL/h.

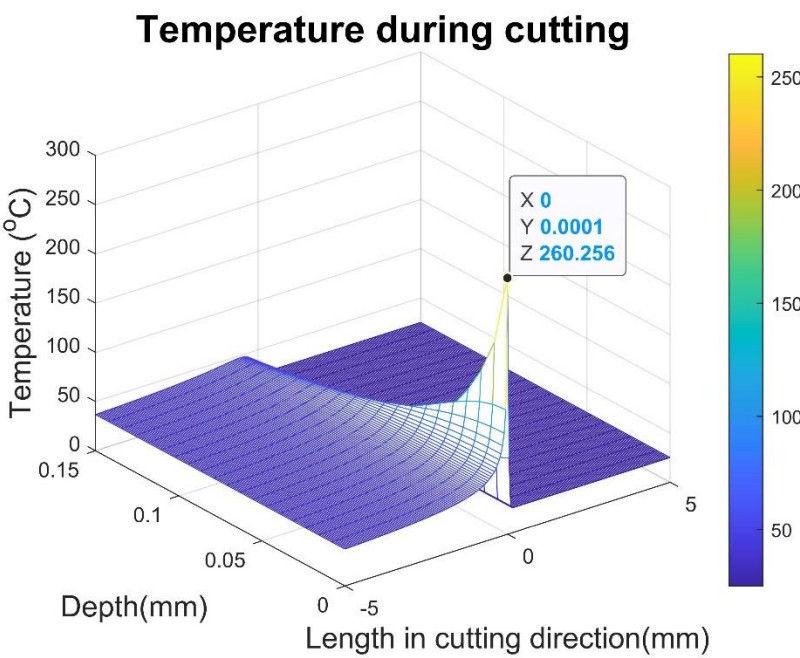

**Figure 8.** Predicted temperature field with feed rate of 200 mm/min and depth of cut of 0.5 mm under dry conditions.

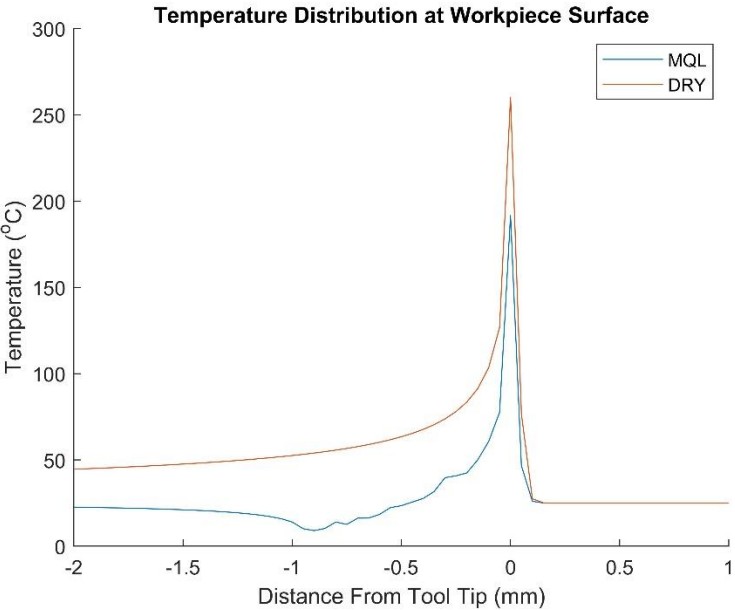

**Figure 9.** Predicted temperature distribution comparison between predicted result at the workpiece surface.

In the proposed model, three heat change sources are considered. The influence of each heat source is evaluated. Figure 10 shows the predicted heat source and heat loss under both MQL and dry conditions for the same process parameters as in Figures 7–9. With the same cutting parameters applied, the heat generated in the primary heat zone for both cases does not show a significant difference as in Figure 10a. This implies similar shear effects for both dry and MQL cases. However, the application of MQL leads to a significant decrease in the heat generated from the secondary heat source, where the tool rubs the workpiece. This is caused by the change in friction coefficient due to the applied lubricant. In the dry case, the calculated friction coefficient is 0.6164. However, in the MQL case, the calculated friction coefficient is 0.2322. The application of MQL does not change

the calculated contact length CA. For both cases, CA is 0.0454 mm. This is similar to the experimental results reported by Rahim et al. [31], where the contact length between the dry and MQL case shows little difference. For MQL case, there is an additional heat loss that needs to be taken away from the temperature calculation as in Equation (29). Figure 10c shows the heat loss calculated. The heat loss is most severe at the point where the lubricant is applied. In the dry calculation, there is no heat loss considered. In these two cases, the cutting forces for the MQL and dry cases are 78.8 N and 87.2 N, respectively. The force calculation does not show a significant difference in this case, which is consistent with the conclusion given by Zhang et al. [14]. Based on the current model, the application of MQL mainly affects the heat generated in the secondary heat source because of the lowered friction coefficient as well as the additional heat loss.

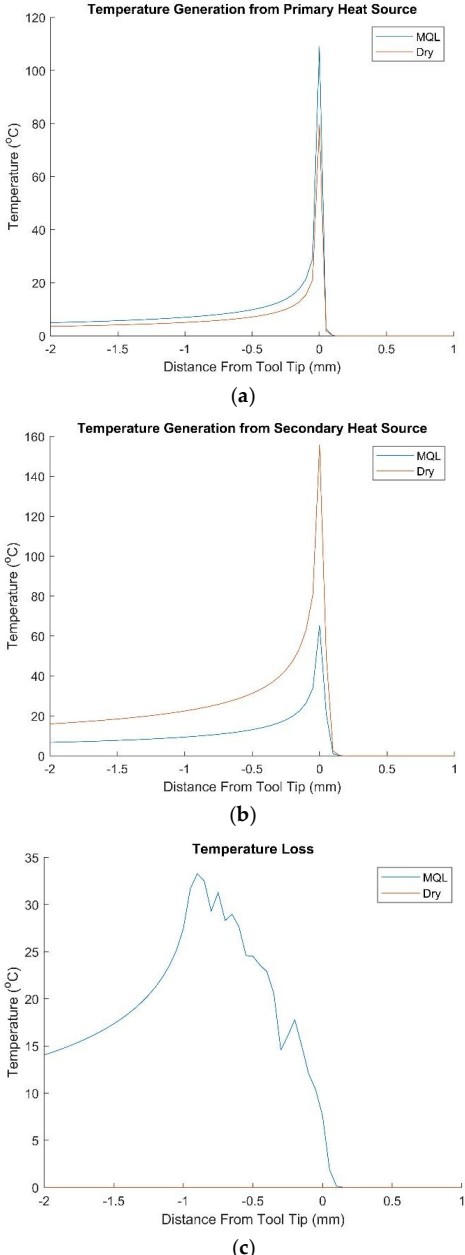

**Figure 10.** Temperature predicted from (**a**) primary heat source and (**b**) secondary heat source. (**c**) Heat loss due to lubrication.

### 4.2. Effect of Feed Rate on MQL Condition

Figure 11 shows the predicted temperature distribution at the workpiece surface around the cutting zone with different applied feed rates (100 mm/min, 150 mm/min, and 200 mm/min). The other parameters are kept the same with the depth of cut of 0.5 mm and flow rate of 400 mL/h. The maximum temperatures calculated are 104.72 °C, 132.78 °C, and 158.55 °C with respect to feed rates of 100 mm/min, 150 mm/min, and 200 mm/min, respectively. This is consistent with the experimentally measured trend, as shown in Figure 12 [29]. However, this trend may not be consistent across different experimental conditions. In a milling experiment performed with AISI 1040 [32], the measured temperature showed a decreasing temperature with a higher feed rate at low cutting speed, while an increasing temperature was observed with a higher feed rate at a high cutting speed. The difference in trends could be led by the difference in material properties. A further study could be conducted to evaluate.

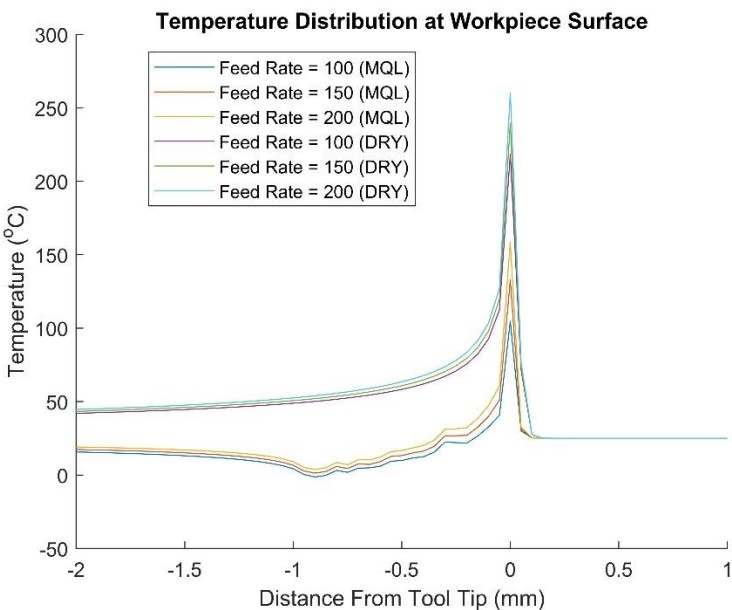

**Figure 11.** Temperature distribution with flow rate of 400 mL/h and depth of cut of 0.5 mm with different feed rates.

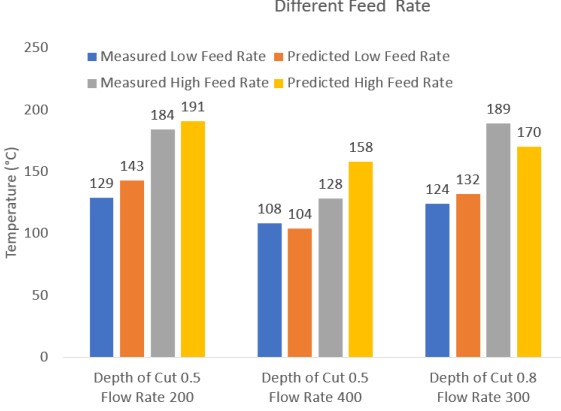

**Figure 12.** Temperature comparison between the low feed rate and the high feed rate.

One of the possible reasons for the increasing temperature under higher feed rate is the increase in the friction coefficient. Based on the proposed model, the calculated friction coefficient is calculated as linearly increasing with increasing feed rate, as shown in Figure 13a. The calculated contact length also shows a similar linear increasing with higher

feed rate, as shown in Figure 13b. The increases in the calculated friction coefficient and contact length both reflect the larger force generated between the tool and the workpiece due to the increased feed rate. The increase in the feed rate means that more materials are removed at a unit time, which requires more forces and further enlarges the effect of rubbing. These effects finally lead to the temperature increase.

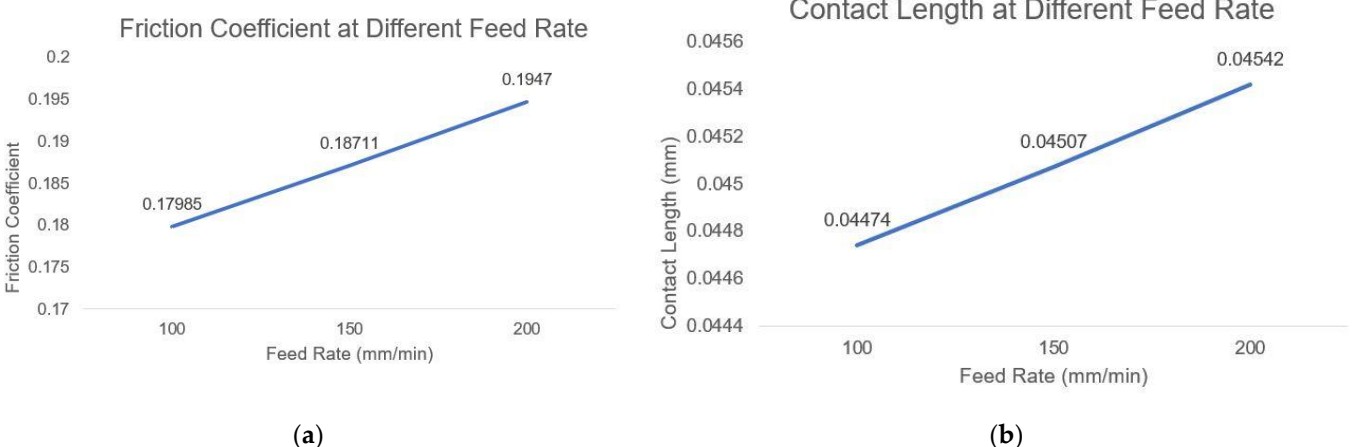

**Figure 13.** (**a**) Friction coefficient and (**b**) contact length calculated under different feed rates with flow rate of 400 mL/h and depth of cut of 0.5 mm.

It is worth mentioning that there is research that shows different trends of the friction coefficient. Banerjee and Sharma [33] performed an analysis of the friction coefficient based on turning operation results on AISI 1045. After minimizing the error between the cutting force, they gained a friction model that is nonlinear with respect to cutting speed and linearly related to the feed rate. However, their results showed that the friction coefficient decreased linearly with the feed rate. As stated before, it could be because of the difference in the experimental setting, as well as the difference between the milling and turning operation.

### 4.3. Effect of Flow Rate on MQL Condition

With the feed rate of 200 mm/min and depth of cut of 0.5 mm fixed, Figure 14 gives the predicted results for temperature field at the workpiece surface under different flow rates, 100 mL/h, 200 mL/h, 300 mL/h, 400 mL/h, and 500 mL/h. The corresponding result for the dry condition is also shown in Figure 14. All the MQL cases show a similar temperature drop and faster convergence as explained in the previous section. To have a better examination, Figure 15a shows the predicted maximum temperature with respect to different lubrication flow rates. With the application of the MQL, the predicted maximum temperature is always lower than that under the dry condition. It is intuitively reasonable because of the lubrication effect. While the higher MQL flow rate further smoothens the rubbing surface between the tool and the workpiece, a higher air flow also leads to more heat dispersion at the flank surface. These two effects combined give a higher temperature drop at the workpiece surface. Under this proposed cutting condition, with increasing flow rate of the lubricants, the maximum temperature first decreases and then increases as the flow rate passes 300 mL/h. The temperature calculated behaves this way mainly because the friction coefficients calculated behave similarly, as shown in Figure 15b. From this point of view, there is a theoretical optimum flow rate for MQL, which leads to a minimum temperature in milling operation. Under this set of cutting conditions, the optimum flow rate is 300 mL/h. The turning experiment performed by Rahim and Dorairaju [34] showed similar effects. In their experiments performed with AISI 1045, with the other parameters kept the same, the measured temperature also decreased first and then increased with higher MQL air pressure, which resulted in the increase in fluid flow.

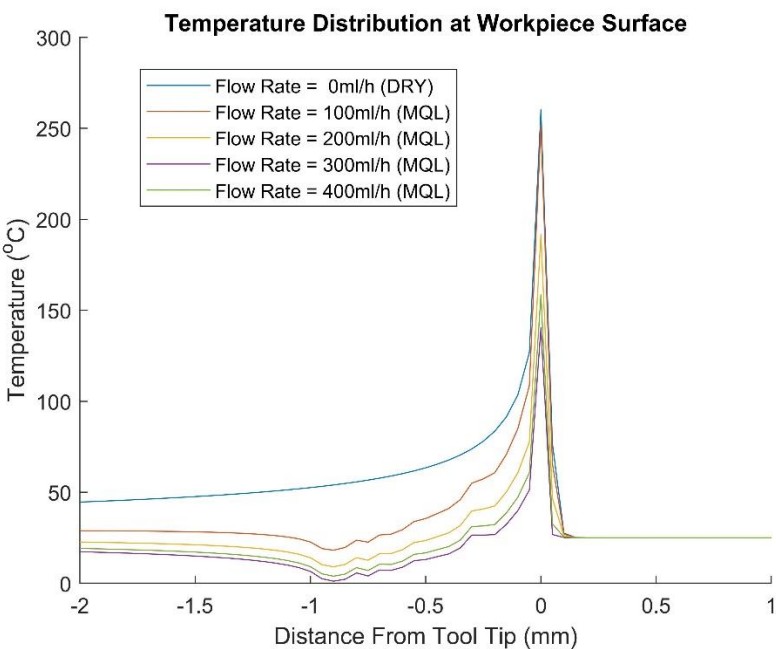

**Figure 14.** Temperature distribution with feed rate of 200 mm/min and depth of cut of 0.5 mm under different flow rates.

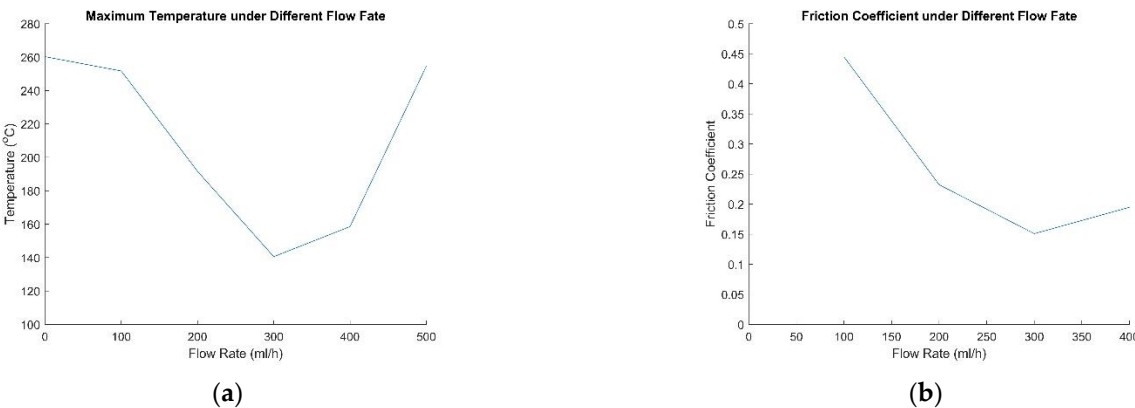

(**a**)                                                                                                              (**b**)

**Figure 15.** Calculated (**a**) maximum temperature and (**b**) friction coefficient with feed rate of 200 mL/min and depth of cut of 0.5 mm under different flow rates.

It is worth mentioning that as the model applies the boundary lubrication condition to predict the friction coefficient change due to MQL, there is an assumption that the lubricants only form a very thin film at the flank surface of the workpiece. However, as the flow rate increases, the lubrication film may overcome the assumed maximum thickness of the boundary lubrication condition. This will cause this model to fail to predict the temperature. This may also explain the large error from the prediction result to the experimental result in Table 5. However, for most MQL applications, the flow rate is less than 200 mL/h [4], so the proposed model should still be reasonable to use for most cases.

### 4.4. Effect of Depth of Cut on MQL Condition

With a feed rate of 150 mm/min and flow rate of 200 mL/h, Figure 16 shows the predicted temperature distribution at the workpiece surface around the cutting zone with different cuts of depth (0.8 mm, 0.5 mm, and 0.2 mm). The corresponding temperature distribution from the dry condition is also shown for comparison. As expected, a large depth of cut leads to a higher temperature in both MQL and dry cases. This increase in temperature with the depth of cut is well recognized by the literature [35,36]. While a larger

depth of cut is performed, it requires a larger cutting force, as shown in Figure 17a, and creates more heat around the cutting zone. The proposed model suggests that as the depth of cut increases, the rate of temperature increase increases as well. This is because the friction coefficient is not linearly correlated with the depth of cut. As shown in Figure 17b, when the depth of cut increases, the friction coefficient increases faster. From a physics point of view, this is because more forces are required by the increase in the depth of cut. The larger force "presses" the tool to be more attached to the workpiece and the chip. This results in both more rubbing between the tool and the workpiece and a higher friction coefficient calculated in this case. The interaction between force and friction is tighter with a higher depth of cut applied.

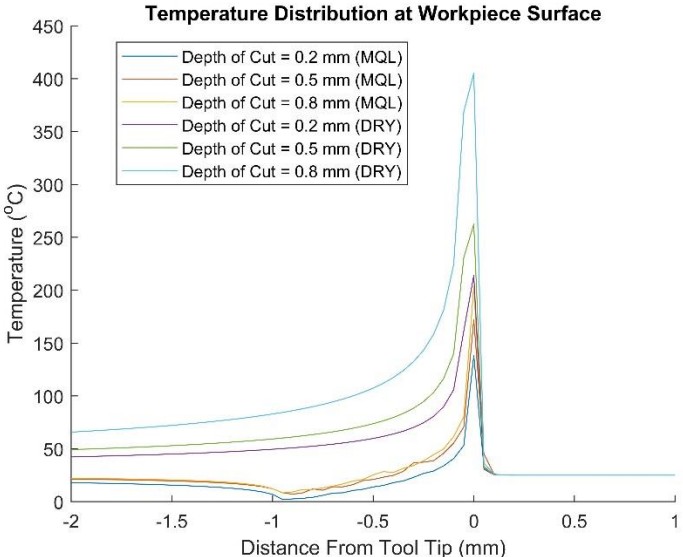

**Figure 16.** Temperature distribution with feed rate of 150 mm/min and flow rate of 200 mL/h under different depths of cut.

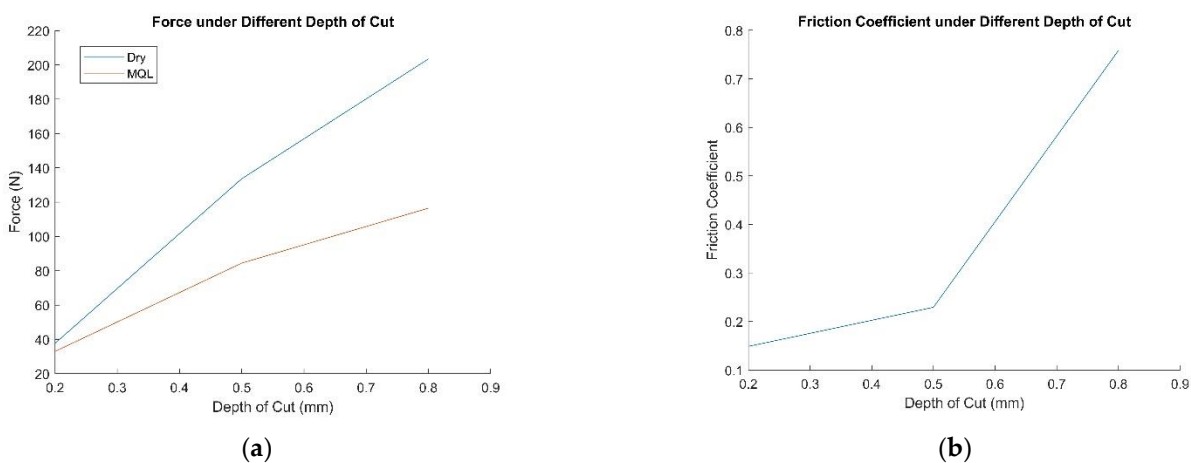

(**a**)                    (**b**)

**Figure 17.** Calculated (**a**) total force and (**b**) friction coefficient with feed rate of 150 mm/min and flow rate of 200 mL/h under different depths of cut.

### 4.5. Other Effects in the Model

Thus far, the proposed model has successfully chased the trend of maximum temperature change due to changes in cutting parameters such as the feed rate, depth of cut, and flow rate of the lubricants. However, several questions remain for further discussion and future work.

First, the original temperature prediction model is adapted from orthogonal cutting model. While the proposed model transfers 3D milling operation into 2D orthogonal cutting,

the choice of rotational angle affects the temperature prediction result. Figure 18 shows the effect of the rotation angle choice on the prediction results with a feed rate of 200 mm/min, depth of cut of 0.5 mm, and flow rate of 400 mL/h. As the angle chosen for prediction changes, the predicted maximum temperature is affected. Feng et al. [37] investigated a similar model for temperature prediction without MQL applied and concluded that the maximum temperature occurs when the rotation angle is chosen between 50° and 70°. Here, in this context, the maximum temperature calculated increases with the rotational angle chosen.

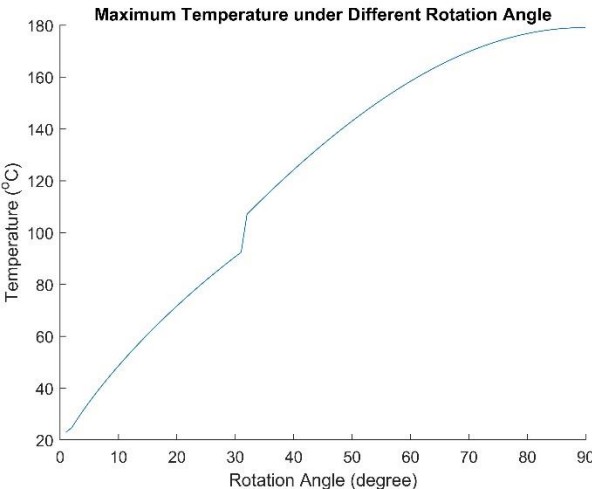

**Figure 18.** Maximum temperature predicted with different choice of rotational angle.

Another factor is the choice of assumed lubricant layer thickness. In the model proposed, the layer thickness is assumed to be linearly correlated to the lubricant flow rate, which is proven useful in the current temperature prediction. However, the accurate formulation requires more research and experiments to verify. While considering the lubrication effect of MQL, the proposed model assumes an average heat transfer coefficient across the workpiece surface, which also needs further experiments for validation.

While the milling operation is a continuous process, the proposed model breaks it into single point cutting at each rotation angle. There could be further effects to be considered as the cutting surface is still 3D in practice. The 3D flow of the lubricants during the tool rotation could also be added into the model for future development, such as the involvement of nanoparticles that provides a better heat dispersion effect [38].

## 5. Conclusions

In this paper, an analytical model is first proposed for the prediction of milling temperature with minimum quantity lubrication. It fills the gap between the analytical modeling and milling temperature prediction, and gives a reasonable temperature prediction within a short time.

In the proposed model, the 3D milling operation is transferred into 2D orthogonal cutting. The modified Oxley's orthogonal cutting model is applied first to have an estimated force, which is further used in the boundary layer lubrication model for an updated friction coefficient calculation. The calculated friction coefficient is then transferred into friction angle in the orthogonal cutting model for further force and temperature calculation. Three sources of heat changes are considered in the proposed model: the primary source due to shearing, the secondary source due to rubbing, and the heat loss at the flank surface due to the applied lubricant and air flow.

The proposed model is then validated with experimental data on milling AISI D2. With different combinations of the cutting parameters, the model-predicted maximum temperatures are compared to the experimentally measured ones. The average absolute error is 10.38%. For each case, the calculation times are all below 1 min. While the proposed

model presents a reasonable maximum temperature prediction, it also catches the trend of temperature changes due to various cutting parameter changes. The following conclusions are drawn:

- With the cutting condition presented, the application of the MQL decreases the cutting temperature at the cutting zone. One mechanism behind the temperature drop is the lowered friction coefficient. With the lower friction coefficient, the heat generated in secondary heat source, with the rubbing between the tool and workpiece, is significantly decreased. Another considered factor is the extra heat loss at the flank surface at the workpiece, which further lowers the temperature.
- With the application of the MQL, the temperature at the flank workpiece surface drops much faster compared to those in the dry cases. This is because of the additional heat loss led by the air flow. At the point where the lubricant is applied, this temperature drop is most obvious.
- With increasing feed rate, the maximum temperature at the cutting zone increases. The higher feed rate requires a higher cutting force, which leads to more rubbing between the tool and the workpiece. The friction coefficient therefore increases with the feed rate. A similar effect occurs with increasing depth of cut.
- The friction coefficient first drops and then increases with flow rate. One possible explanation is that the lubricant has already formed a thick film due to a high lubricant flow rate. As more lubricant is applied, the marginal lubrication effect is decreasing.

Thus far in the literature, the cutting temperature during the milling operation with the MQL condition has not been fully investigated. This proposed model delivered reasonable temperature prediction results within a short calculation time, although limitations existed as the parameter choice needed more calibration and research. The proposed model can still be used as a reference for future temperature prediction.

**Author Contributions:** Conceptualization, L.C. and S.Y.L.; investigation, Y.-T.L., Y.-F.L. and T.-P.H.; resources, F.-C.H.; data curation, Y.-T.L., Y.-F.L. and T.-P.H.; writing—original draft preparation, L.C.; writing—review and editing, Y.F.; supervision, S.Y.L.; project administration, F.-C.H.; funding acquisition, S.Y.L. All authors have read and agreed to the published version of the manuscript.

**Funding:** This research received no external funding.

**Data Availability Statement:** Not applicable.

**Acknowledgments:** Thanks to The Metal Industries Research and Development Centre from Taiwan for funding this research and providing help.

**Conflicts of Interest:** The authors declare no conflict of interest.

## Nomenclature

| | |
|---|---|
| $a_{wk}$ | The thermal diffusivity of the workpiece |
| $A_{bs}$ | The adsorbed lubricant film contact area |
| $A_{ms}$ | The metallic contact area |
| $C^*$ | The side cutting edge angle |
| $C_s$ | The tool side cutting angle |
| $C_t$ | The density of the workpiece |
| $C_{wk}$ | The thermal capacity of the workpiece |
| $CA$ | The contact length |
| $d$ | The axial depth of cutting |
| $D$ | The inclination distribution function |
| $F_c$ | The cutting force |
| $F_t$ | The tangential force |
| $K_0$ | The modified Bessel function |

| | |
|---|---|
| $k_t$, | The thermal conductivity of the tool |
| $k_{wk}$ | The thermal conductivity of the workpiece |
| $L$ | The length of the shear plane |
| $L_{eff}$ | The effective lubricated length |
| $\bar{h}$ | The average heat transfer coefficient |
| $H_{\max}$ | The asperity height distribution |
| $n_0$ | The total asperity number |
| $N$ | The normal load |
| $i$ | The inclination angle |
| $i^*$ | The equivalent inclination angle |
| $k_{air}$ | The thermal conductivity of the air |
| $L_{eff}$ | The effective lubricated length |
| $p_b$ | The yield pressure at the adsorbed lubricant film contact area |
| $p_m$ | The yield pressure at the metallic contact area |
| $Pr$ | The Prandtl number |
| $q_{shear}$ | The heat source density in the shear zone |
| $q_{rub}$ | The heat source density in the rubbing zone |
| $r$ | The asperity tip radius |
| $R$ | Cutter radius |
| $Re$ | The Reynolds number |
| $s_b$ | The shear strength at the adsorbed lubricant film contact area |
| $s_m$ | The shear strength at the metallic contact area |
| $t_1$ | The undeformed chip thickness |
| $t_b$ | The effective adsorbed lubrication film thickness |
| $t_c$ | Depth of cut |
| $\bar{t}_c$ | Average depth of cut |
| $t_c^*$ | The equivalent orthogonal cutting |
| $T_{flank}$ | The average tool flank face temperature |
| $T_{primary}$ | The temperature rise in the workpiece from the primary heat source |
| $T_{Secondary}$ | The temperature rise in the workpiece from the secondary heat source |
| $T_{loss}$ | The temperature drop in the workpiece from the heat loss |
| $T_w$ | The room temperature |
| $V$ | Cutting speed |
| $V_f$ | Feed rate |
| $V_r$ | The rotation speed |
| $V(\psi)$ | The equivalent cutting speed |
| $w$ | The cutting width |
| $w^*$ | The equivalent cutting width |
| $\alpha$ | The rake angle |
| $\alpha^*$ | The equivalent rake angle |
| $\eta_c$ | The chip flow angle |
| $c^*$ | The equivalent chip flow angle |
| $\gamma$ | The heat distribution coefficient |
| $\rho_t$ | The density of the tool |
| $\rho_{wk}$ | The density of the workpiece |
| $\mu$ | The friction coefficient |
| $\psi$ | Tool rotation angle |
| $\phi$ | Shear angle |

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
