# Peer review of "Analytical Model for Temperature Prediction in Milling AISI D2 with Minimum Quantity Lubrication"

_metals, doi:10.3390/met12040697_

Round 1

Reviewer 1 Report

The reviewer comments of the paper «Analytical Prediction of Milling Temperature with Minimum Quantity Lubrication»- Reviewer

The authors presented an article «Analytical Prediction of Milling Temperature with Minimum Quantity Lubrication». However, there are several points in the article that require further explanation.

Comment 1:

Title needs to be concretized. Add stock material in the title. Add the "FEM" method to the title.

The abstract needs to be improved.

Demonstrate in the abstract novelty, practical significance. Add quantitative and qualitative work results to the abstract. However, add an method error. Describe the input and output parameters investigated in the work. It is necessary to describe the most results of the article.

Comment 2:

The introduction needs to be improved.

Now the list of references needs to be supplemented with at least 8-10 more references published over the past 5 years. Refer to the works of the authors Gupta M.K., Machado A., Pimenov D.Y., etc. Here are some recent articles:

Metals 2021, 11, 1674. doi:10.3390/met11111674

Materials 2021, 14, 7207. doi:10.3390/ma14237207

Industrial Lubrication and Tribology 2019, 71(2), 267-277. doi:10.1108/ILT-11-2017-0322

It is necessary to add a paragraph and a detailed analysis of the studied material AISI D2 steel of the workpiece. What difficulties are there in the machining and milling process especially? Why is this material so important?

After analyzing the literature, show before formulating the goal of the "blank" spots. Which has not been previously done by other researchers. You must show the importance of the research being undertaken. Show what will be the new research approach in this article. You need to show a hypothesis.

Demonstrate in the abstract novelty, practical significance.

Add a clear purpose to the article.

Comment 3:

  1. Analytical Model for Temperature Prediction in Milling

Are all figures original? If not needed appropriate citations and permissions. Refine this for figures throughout the article.

What is the hardness of the workpiece and how was it measured?

Describe in table the geometry of the cutter used in the research (diameter, number of teeth, rake and clearance angle, etc.). Show these dimensions in the photo.

Show the direction of the machine axes. How does this compare to measured cutting forces? What kind of milling scheme is used? Describe in the text.

Describe the measurement procedure in more detail. At what point in time? How is the measuring setup set up? How many repetitions of measurements? What statistical methods are used to process experimental results? Describe the experimental stand in more detail. What method of experiment planning is used and why?

Comment 4:

Finite Element modeling should be described and illustrated in more detail.

Add a design diagram with boundary conditions on the shape. Justify your choice and assumptions made. What type of finite elements are used and why? How does the size of finite elements affect the accuracy of calculations? What parameters did you calculate using the FEM and why?

Give the parameters of the PC on which the forces were calculated using the FEM. How much time was spent on calculations? It is useful to give such parameters in the table and briefly explain the performance in the text.

Comment 5:

  1. Model Analysis

The description of all figures in the text must be supplemented. Minimum 4-5 sentences. It is also important to add a figure with output curves from cutting data. Analyze the nature of these curves in accordance with the influence of the cutting mode on these curves, feed, cutting speed, depth of cut. This needs to be explained in terms of cutting physics. What is the difference from previous work in this area?

Is it important to show how the validation of the FEM models was assessed?

Comment 6:

It will be useful to add a section of Nomenclature in which to sign all the physical quantities and abbreviations encountered in the article. There are many physical quantities in the text and such a section will help to find the description of the necessary element.

For example,

  • : Density (g/cm3)

FEM         : Finite Element Modelling

etc.

Comment 7:

Conclusions.

It is necessary to more clearly show the novelty of the article and the advantages of the proposed method. Add qualitative and quantitative results of your work. What is the error of the obtained models? What is the difference from previous work in this area? Show practical relevance.

Comment 8

The quality and resolution of all figures needs to be improved.

Comment 9:

Proofreading by a native English speaker is required.

The article is interesting, but needs to be improved. Authors should carefully study the comments and make improvements to the article step by step. After major changes can an article be considered for publication in the "Metals".

Reviewer 2 Report

The article presents an analytical study on milling temperature with MQL. The analytical aspect of MQL is not much investigated; thus, there is a kind of novelty in this work. However, sufficient changes need to be made to improve the manuscript's quality.  

  1. Citing the reference of  Le Coz et al. [6]  that little research has been conducted to the milling temperature of milling is a decade old, is not appropriate in support of this investigation. Several articles have appeared in the last ten years. A through litrature reivew on MQL must be provided.
  2. The model is validated with available data in the Khan et al. The work of Khan et al. does not provide sufficient information to validate the work. How are the following are obtained for validation, which are not given in Khan et al. - total asperity number, inclination distribution function, approach of two surfaces,  asperity height distribution.
  3. In the absence of those mentioned above, the work can't be considered validated. Provide the values used/assumed for the above stated. The authors must refer to the available model for prediction of the MQL friction coefficient to benchmark their results, see the article on, Identification of a friction model for minimum quantity lubrication machining https://doi.org/10.1016/j.jclepro.2014.07.034
  4. The data does not give the face milling tool details ( single or multi-point cutting ). At the same time, the current article assumes a single-point cutting tool. The mismatch needs to be mentioned as a limitation of the article. 
  5. The lubrication film model considered in this article is for pure fluid. While the validation data is that of nanofluid. The possible impact of using a base fluid in place of nanofluid should be mentioned with appropriate reference, see https://doi.org/10.1016/j.jclepro.2020.122553
  6. The error calculated is 'absolute error' and not 'error.'
  7. "Except that, the proposed model gives a reasonable temperature prediction result compared to the experimental data. Further analysis has been done to the model in Section 4," which is misleading. Need correction.
  8. Results of sections 4.2, 4.3,  and 4.4 need to be compared (in terms of rage, trend, and order) with a similar investigation where the attributes are measured.  
  9. The simulation results are merely presented vis-a-vis graphs are described in words. The scientific discussion is missing. The results must be explained based on the physics of the process.
  10. Overall language quality of the article is below acceptance. A thorough revision is required. 

Round 2

Reviewer 1 Report

Now the article can be published.

Reviewer 2 Report

The changes made in the revised manuscript have improved the quality of the manuscript.